🔓 | **Open Peer Review** | Clinical Microbiology | Research Article

# Identification and predictive machine learning models construction of gut microbiota associated with lymph node metastasis in colorectal cancer

Yongzhi Wu,[1] Chengen Deng,[2] Zigui Huang,[1] Yongqi Huang,[1] Chuanbin Chen,[1] Mingjian Qin,[1] Zhen Wang,[1] Fuhai He,[1] Shenghai Liu,[1] Rumao Zhong,[1] Jun Liu,[1] Chenyan Long,[1] Jungang Liu,[1] Weizhong Tang,[1] Xiaoliang Huang[1]

**ABSTRACT**  This study focuses on the significant role between gut microbiota and lymph node metastasis (LNM) in colorectal cancer (CRC). By conducting 16S rRNA sequencing on fecal samples from 147 CRC patients and combining it with the linear discriminant analysis effect size algorithm, we successfully identified significant differences in the gut microbiota between patients with LNM and those with no lymph node metastasis (NLNM). Furthermore, using transcriptome data from 23 CRC patients, we constructed an immune cell infiltration matrix to deeply explore the biological functions associated with LNM. Eventually, using the characteristics of the gut microbiota associated with LNM, we developed random forest (RF) and multilayer perceptron (MLP) machine learning models to predict the LNM status of CRC patients. We identified 21 differentially abundant gut microbes between the two groups, among which *Bacteroides plebeius*, significantly enriched in the LNM group, is closely related to the upregulation of neutrophils and chemokine CXCL8 expression, and this bacterial species is also positively correlated with the enhancement of inosine monophosphate metabolism. The RF and MLP models constructed based on the LNM-associated gut microbiota showed good predictive efficacy in predicting LNM status in CRC. This study reveals that *Bacteroides plebeius* may play an important role in the progression of CRC, with its mechanism potentially involving changes in immune modulation and metabolic pathways. The classification model constructed based on gut microbiota characteristics can predict LNM status of CRC, providing a new perspective for personalized and precision treatment of CRC patients.

**IMPORTANCE**  This study highlights the pivotal role of gut microbiota in lymph node metastasis (LNM) of colorectal cancer (CRC), identifying key microbial differences between LNM and NLNM groups. Our findings implicate *Bacteroides plebeius* in CRC progression via immune modulation and metabolic alterations. Moreover, machine learning models based on gut microbiota predict LNM status accurately, offering a novel approach for personalized CRC treatment.

**KEYWORDS**  colorectal cancer, lymph node metastasis, intestinal microbiology, 16S rRNA, machine-learning

Colorectal cancer (CRC) ranks as the third most common malignant tumor and the second leading cause of cancer-related deaths worldwide, with continued attention on its prevention and treatment (1). However, there are still over 1.9 million new cases and 900,000 deaths annually. Despite continuous advancements in diagnostic and therapeutic technologies, treatment-related adverse effects and the poor prognosis associated with the disease—characterized by high rates of local recurrence and distant

**Peer Reviewer** Shi Huang, The University of Hong Kong, Hong Kong, Hong Kong

Address correspondence to Xiaoliang Huang, xiaoliang@outlook.com, Weizhong Tang, tangweizhong@gxmu.edu.cn, or Jungang Liu, liujungang@gxmu.edu.cn.

Yongzhi Wu, Chengen Deng, and Zigui Huang contributed equally to this article. The order of authorship was fully discussed and unanimously agreed upon by all authors and is based primarily on the substantive contributions of each author to this research.

The authors declare no conflict of interest.

See the funding table on p. 19.

metastasis (2)—contribute to the persistently high mortality rate of CRC, imposing a substantial burden on both patients and society. Moreover, a growing number of CRC patients are diagnosed with lymph node metastasis (LNM) at initial presentation, and the presence of LNM significantly reduces the 5-year survival rate (3), serving as a critical negative prognostic factor in clinical outcomes.

Lymph node involvement has significant guiding implications for the treatment and prognosis of colorectal cancer; a positive status can identify patients who need adjuvant chemotherapy (4), and a negative status is also an important independent factor for the prognosis of patients with high microsatellite instability (5). Emerging evidence suggests that LNM and distant metastasis may share a common clonal origin (6), underscoring the critical importance of early LNM detection for improving overall prognosis. Currently, LNM diagnosis predominantly relies on postoperative pathological evaluation, while effective preoperative predictive tools remain scarce—a limitation that severely restricts the implementation of precision treatment strategies. Recent studies have attempted to enhance diagnostic accuracy through multimodal data integration. For instance, Xia et al. developed a magnetic resonance imaging (MRI)-pathology fusion model for rectal cancer lymph node assessment (7); however, its generalizability still needs to be verified. In contrast, machine learning (ML) approaches, leveraging high-dimensional biomarker data, show promise in achieving personalized LNM risk stratification (8). Consequently, elucidating the molecular mechanisms underlying LNM and developing non-invasive biomarker-driven predictive models have emerged as pivotal research directions to optimize CRC clinical management.

The role of gut microbiota in the maintenance of intestinal homeostasis provides an important basis for the theory of microorganisms as environmental triggers of CRC (9). Advances in 16S rRNA sequencing, metagenomics, germ-free murine models, and fecal microbiota transplantation (FMT) have propelled microbiome research into a mechanistic era (10), enabling the identification of CRC-associated oncogenic taxa such as *Fusobacterium nucleatum* (*Fn*), colibactin-producing *Escherichia coli* (pks+ *E. coli*), and enterotoxigenic *Bacteroides fragilis* (ETBF). These pathobionts drive carcinogenesis and metastasis through multifaceted mechanisms, including immunosuppressive microenvironment remodeling (11, 12), genotoxin production (13), and metabolic reprogramming (14). Notably, *Fn* is enriched in tumor tissues of CRC patients with LNM (15), and preclinical studies demonstrate that *Fn*-pretreated HCT-116 cells exhibit enhanced invasiveness, leading to increased pulmonary metastasis in nude mice (16). Despite these advances, the host-microbiota crosstalk during metastatic progression—particularly in LNM—remains poorly understood. Furthermore, while fecal microbiota profiling offers non-invasive diagnostic potential, the lack of validated LNM-specific microbial signatures and functional pathways severely hinders clinical translation.

In this study, 147 CRC patients were included, preoperative stool samples were collected for 16S rRNA sequencing, and 23 paired tumor tissues were selected for transcriptome sequencing, aiming at identifying the differential gut microbiota in patients with LNM and no lymph node metastasis (NLNM), and integrating transcriptome data to resolve the correlation between flora-immune microenvironment interactions, and finally combining the characteristic microbiomes to construct a random forest (RF) and multilayer perceptron (MLP) machine learning models in conjunction with the characteristic microbiome to achieve non-invasive prediction of LNM status. This study combines multi-omics data with ML algorithms to systematically reveal the molecular mechanisms of gut microbiota driving CRC-LNM through immune-metabolic regulation and biological pathways and to provide novel biomarkers for individualized therapeutic decision-making.

## RESULTS

### Clinical data and statistical characteristics of CRC patients enrolled in the study

Researchers collected 198 fecal samples from 236 CRC patients who met the study criteria for 16S rRNA sequencing analysis. A total of 147 patients with complete LNM status data were selected. They were then divided into two groups based on lymph node negativity and positivity: the NLNM group with 69 individuals and the LNM group with 78 individuals. During the recruitment phase, we adopted a consecutive enrollment strategy without active matching for known LNM risk factors (e.g., age, gender, tumor subtype, or localization) to preserve the heterogeneity of real-world clinical populations. Despite this non-selective approach, baseline characteristics—including age ($P = 0.880$), gender ($P = 0.053$), tumor localization ($P = 0.187$), tumor volumes ($P = 0.903$), and mismatch repair (MMR) status ($P = 0.568$)—showed no statistically significant differences between the LNM and NLNM groups (all $P > 0.05$; Table 1). This natural equilibrium in potential confounders validates the comparability of the cohorts, minimizing selection bias and reinforcing the robustness of subsequent gut microbiota analyses.

### Comparison of microbiome diversity between LNM and NLNM groups

In order to assess potential differences in gut microbiome diversity between LNM and NLNM groups, we analyzed microbial community diversity and structure using 16S rRNA sequencing. As shown in Fig. 1A, none of the six α-diversity indices (observed, Chao1, ACE, Shannon, Simpson, coverage) exhibited statistically significant differences between groups ($P > 0.05$ for all), indicating comparable species richness, evenness, and sequencing depth. The β-diversity analysis (Fig. 1B) revealed distinct microbial compositions: the Jaccard index showed significant separation ($P < 0.05$), while Bray-Curtis dissimilarity approached significance ($P = 0.0586$), suggesting divergence primarily in species composition rather than abundance. These findings highlight that while α-diversity remains homogeneous, β-diversity—particularly species composition— serves as a critical discriminator between LNM and NLNM groups' microbiomes. This structural divergence provides a robust foundation for subsequent differential taxa analysis to identify metastasis-associated microbial biomarkers.

**TABLE 1** Demographic and clinical characteristics of CRC patients stratified by positive and negative lymph node metastasis

| Clinical characteristic[a] | Detail | Negative lymph node patients ($n$ = 69) | Positive lymph node patients ($n$ = 78) | $P$-value[b] | Test |
|---|---|---|---|---|---|
| Age (years, mean ± SD) | | 58.32 ± 11.19 | 58.04 ± 11.22 | 0.880 | $t$-test |
| Age (%) | <60 | 41 (59.4) | 40 (51.3) | 0.410 | Pearson $\chi^2$ |
| | ≥60 | 28 (40.6) | 38 (48.7) | | |
| Gender (%) | Male | 48 (69.6) | 41 (52.6) | 0.053 | Pearson $\chi^2$ |
| | Female | 21 (30.4) | 37 (47.4) | | |
| BMI (%) | <24.0 | 42 (60.9) | 57 (73.1) | 0.162 | Pearson $\chi^2$ |
| | ≥24.0 | 27 (39.1) | 21 (26.9) | | |
| Tumor localization (%) | Left colon | 20 (29.0) | 18 (23.1) | 0.187 | Pearson $\chi^2$ |
| | Right colon | 20 (29.0) | 14 (18.0) | | |
| | Rectum | 28 (40.6) | 43 (55.1) | | |
| | Transverse colon | 1 (1.4) | 3 (3.8) | | |
| Tumor volume (cm$^3$, mean ± SD) | | 23.69 ± 18.54 | 34.51 ± 19.18 | 0.903 | $t$-test |
| MMR status | dMMR | 4 (6.0) | 2 (2.6) | 0.568 | Pearson $\chi^2$ |
| | pMMR | 65 (94.0) | 76 (97.4) | | |

[a]BMI, body mass index; dMMR, deficient mismatch repair; pMMR, proficient mismatch repair.
[b]$P$ values <0.05 were statistically significant.

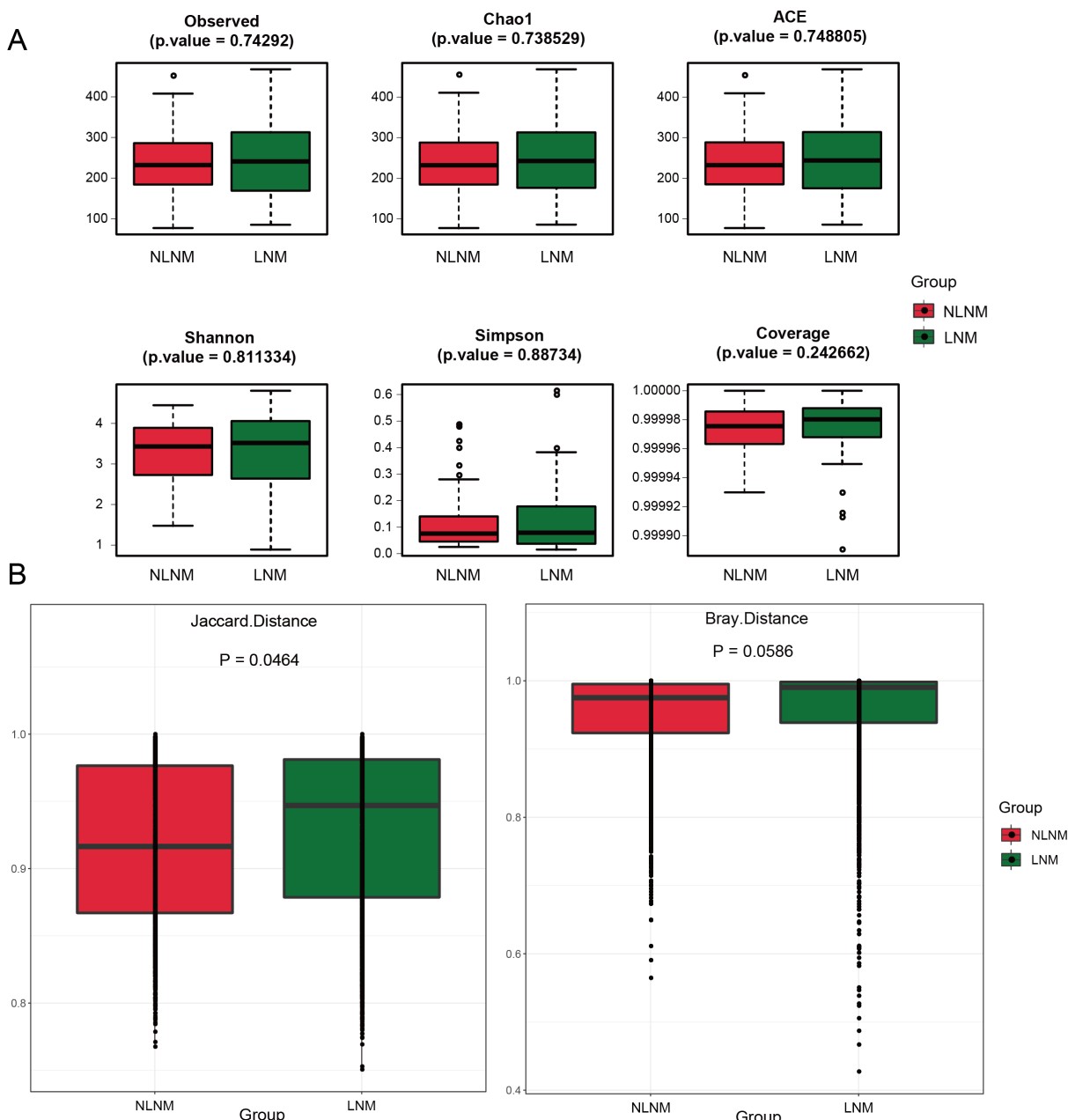

FIG 1 Comparison of microbiome diversity index of CRC patients in LNM and NLNM. (A) Comparison of α-diversity index of gut microbiota between LNM and NLNM of CRC patient group. Diversity analysis was performed based on the Wilcoxon rank sum test, using *P*-value < 0.05 as the significance screening threshold for differences, and used the Bonferroni method of multiple hypothesis testing carried out to verify the *P*-value false discovery rate (FDR), to evaluate whether two groups of species diversity exist significant differences. The figure shows the six α-diversity indices of the two groups of samples. The abscissa represents different groups, and the ordinate represents the community diversity index value of the sample group. (B) Comparison of β-diversity index of gut microbiota between LNM and NLNM. The horizontal axis represents the group, while the vertical axis indicates the community diversity index value for each group of samples.

## Exploration of LNM-associated gut microbiota

By integrating linear discriminant analysis effect size (LEfSe) and microbial correlation networks, we systematically dissected compositional and ecological differences in gut microbiota between LNM and NLNM groups. The bar chart in Fig. 2A compares the absolute abundance of different taxonomic units of gut microbiota between the NLNM and LNM groups, revealing significant differences in species composition. For instance,

*Bacteroides_plebeius* is more abundant in the LNM group, whereas *Clostridium_ramosum* is more prevalent in the NLNM group. On the other hand, Fig. 2B and Table S1 revealed microbial taxonomic units with significant differences between the two groups by LEfSe analysis, with the linear discriminant analysis (LDA) scores on the horizontal axis reflecting the degree of enrichment of these taxonomic units in the corresponding groups, with higher scores being associated with higher levels of enrichment. In addition, we found a total of 21 taxa with statistically significant differences in abundance at different taxonomic levels, of which 12 taxa were significantly more abundant in LNM than in NLNM group. For example, *g__Bacteroides.s__Bacteroides_plebeius* had the highest degree of enrichment in LNM compared to NLNM, which may be closely related to the patient's LNM. To further understand the interactions within the microbial communities of the two groups of patients, we constructed a correlation network diagram of the dominant gut microbiota (Fig. 2C). In this network, green nodes represent microorganisms significantly enriched in LNM group, while red nodes represent microorganisms significantly enriched in NLNM group. The yellow connecting lines indicate positive correlations, and the blue connecting lines indicate negative correlations. Among them, the dominant bacteria in the NLNM group, *o__Actinomycetales.f__Corynebacteriaceae* and *f__Corynebacteriaceae.g__Corynebacterium*, and the dominant bacteria in the LNM group, *g__Vagococcus.s__Vagococcus_teuberi*, were the three bacteria most closely connected to other nodes. This indicates that these three bacteria have the closest correlation with other dominant bacteria. Additionally, we found that *g__Howardella.s__uncultured_organism* in the LNM group showed a significant negative correlation with *o__Actinomycetales.f__Corynebacteriaceae* and *f__Corynebacteriaceae.g__Corynebacterium* in the NLNM group. Based on these results, we speculated that there may be potential competitive relationships between the dominant microbial communities in the two groups.

## Biological functional prediction of gut microbiota in LNM and NLNM groups

To explore the biological pathways enriched by genes in the gut microbiome of colorectal cancer patients under different lymph node states, Phylogenetic Investigation of Communities by Reconstruction of Unobserved States II (PICRUSt2) was used to predict the Kyoto Encyclopedia of Genes and Genomes (KEGG) pathways of LNM and NLNM patients. A total of 174 different KEGG pathways were identified, among which four pathways showed statistically significant differences (Fig. 3; Table S2) ($P < 0.05$). Figure 3A displays the absolute abundance distribution of the four key metabolic pathways between LNM and NLNM groups. Figure 3B further quantifies their mean proportional differences with 95% confidence intervals. Among these, the *Vibrio cholerae* infection pathway was significantly enriched in the LNM cohort ($P = 0.034$), whereas three pathways—renin-angiotensin system ($P = 0.016$), steroid biosynthesis ($P = 0.028$), and endocytosis ($P = 0.033$)—showed marked enrichment in the NLNM group. These suggest that these pathways may be closely associated with the development of LNM. Their high expression in different groups may mirror changes in gut microbiota metabolic functions under varying LNM statuses and also demonstrates remarkable differences in gut microbiota metabolism between the two patient groups.

## Correlation between LNM-related gut microbiota and tumor-infiltrating immune cells

Tumor-infiltrating immune cells play a crucial role in the tumor immune microenvironment, influencing tumor-associated immune responses, thereby inhibiting tumor growth or potentially promoting tumor metastasis and immune evasion (17). Tumor-infiltrating immune cells are potential targets for cancer immunotherapy. Therefore, we utilized transcriptomic sequencing data to investigate the composition of 22 infiltrating immune cells in 23 CRC patients (12 in NLNM group and 11 in LNM group). This included evaluating immune cell abundance differences, analyzing correlations between gut microbiota and immune cells, and constructing an association network. Figure 4A showed

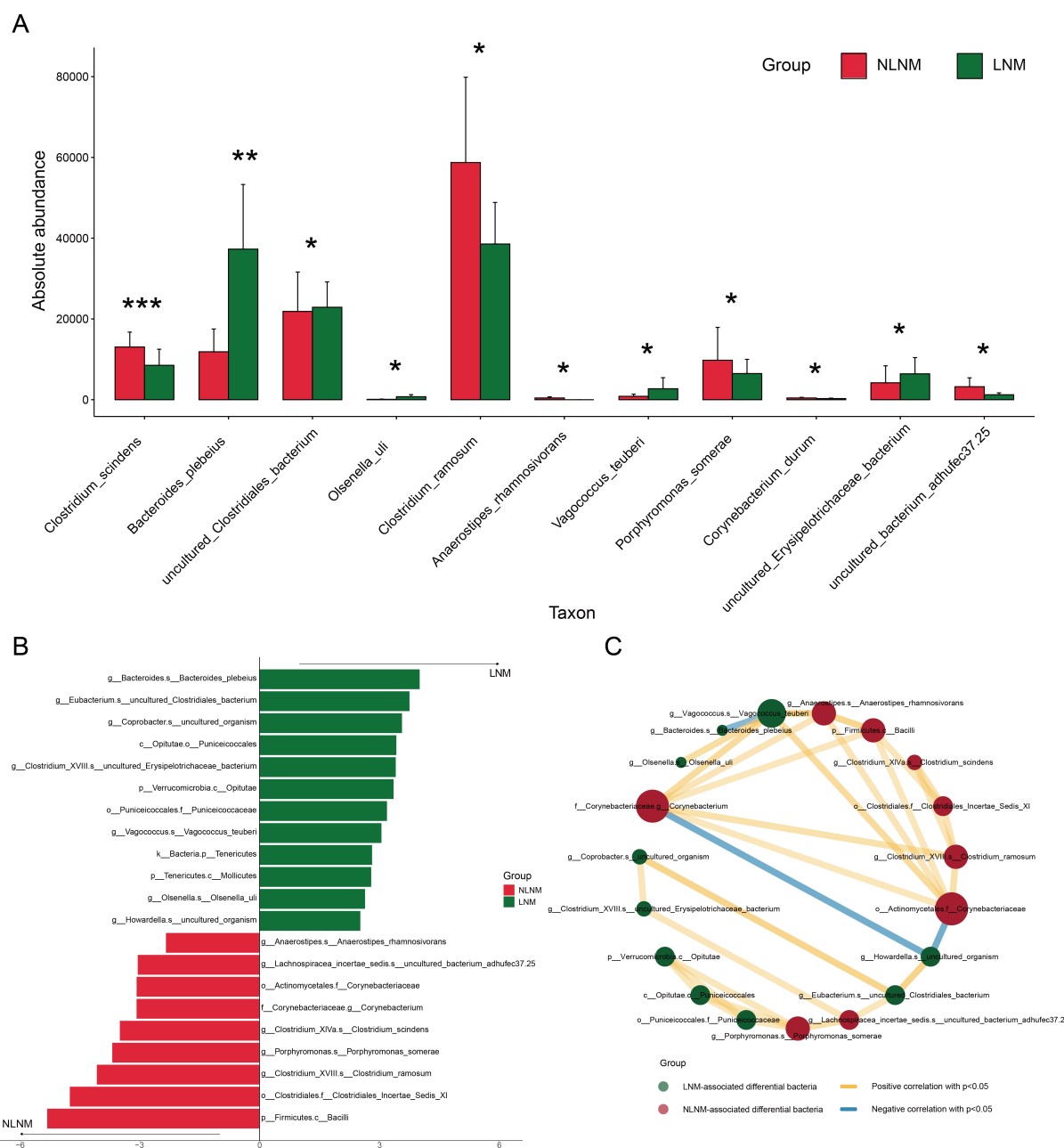

**FIG 2** Analysis of gut microbiota difference between CRC patients in LNM and NLNM. (A) Comparison of absolute abundance of species taxon in two groups of patients. The vertical axis shows the absolute abundance of gut microbiota, ranging from 0 to 80,000. The horizontal axis lists distinct bacterial taxa. For each taxon, two bars represent the absolute abundance in the NLNM and LNM groups. The bar height indicates the mean abundance, and the error bars show the standard deviation or confidence interval. "*" means a statistically significant difference ($P < 0.05$) between the groups, "**" denotes a highly significant difference ($P < 0.01$), and "***" indicates an extremely significant difference ($P < 0.001$). (B) LDA bar graph based on 16S rRNA gene sequencing. The color of the bar graph represents the group, the horizontal coordinate is the LDA score (processed by log10), the vertical coordinate represents the distinct species in the group with significantly higher abundance, and the length of the histogram represents the magnitude of the LDA score value. (C) Network diagram of LNM-related differential gut microbiota correlation. Each node represents each species. The node color represents the group. The node size represents the number of edges connected to the node. The larger the node, the more the number of edges connected to the node. The connection line represents a significant correlation between the two nodes. Spearman phase relationship value less than 0 (negative correlation) represents the blue line, and the number of nodes is the same. A Spearman correlation value greater than 0 (positive correlation) represents a yellow line. The thicker the line, the greater the Spearman correlation coefficient between the two nodes.

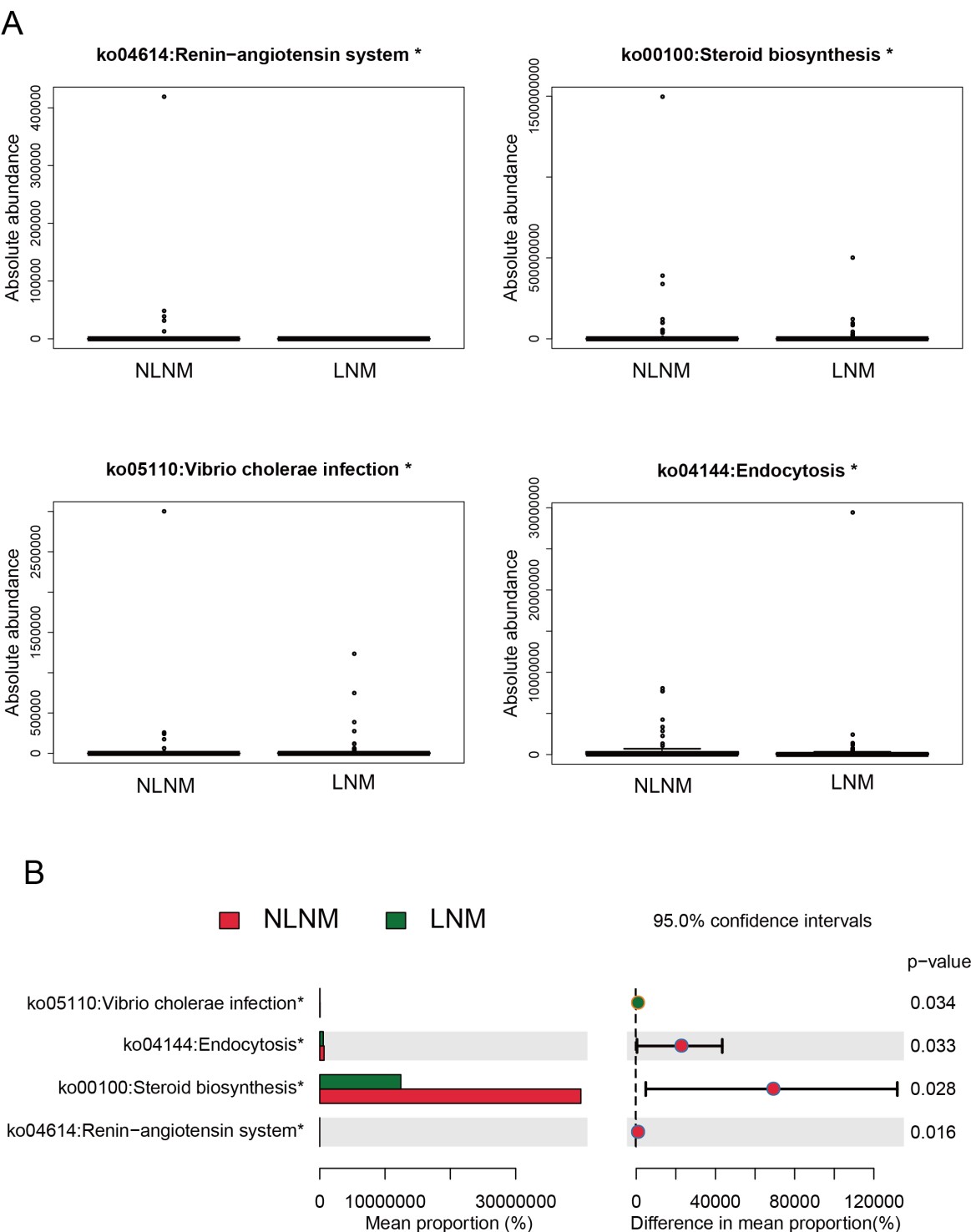

FIG 3 KEGG pathways prediction with significant differences between two groups. (A) Box plot of KEGG functional abundance between LNM and NLNM. The horizontal coordinate represents the grouping, and the vertical coordinate represents the predicted abundance. In each sample, the box edge represents the 25th to 75th percentile, the horizontal black line represents the median, and the extension line is equal to 1.5 times the maximum and minimum values between the quartiles. (B) KEGG pathways prediction histogram with significant differences between LNM and NLNM. The horizontal coordinates of the left panel represent mean proportion, and the horizontal coordinates of the right panel represent difference in mean proportion. The vertical coordinates represent the four KEGG pathways that are significantly different in LNM and NLNM. The "*" in the upper right corner of the KEGG pathway indicates the size of the P-value: "*" for $0.01 \leq P < 0.05$, "**" for $0.001 \leq P < 0.01$, and "***" for $0.0001 \leq P < 0.001$.

abundance differences of various immune cell types between LNM and NLNM groups. CD8+ T cells abundance was lower in LNM than in NLNM ($P = 0.04$). Figure 4B and C showed the correlations between elevated gut microbiota and immune cells in the two groups, respectively. In NLNM group, g__Anaerostipes.s__Anaerostipes_rhamnosivorans was significantly positively correlated with eosinophils, monocytes, and resting dendritic cells. In LNM group, g__Bacteroides.s__Bacteroides_plebeius was significantly positively correlated with neutrophils; g__Olsenella.s__Olsenella_uli was significantly positively correlated with naive B cells and activated mast cells. Figure 4D further illustrated the associations between gut microbiota and immune cells. These findings indicate that there are differences in immune cell infiltration between the LNM and NLNM groups in CRC patients, and that LNM-associated dominant gut microbiota is closely linked to various tumor-infiltrating immune cells. This suggests a potential regulatory role of LNM-associated differential gut microbiota in shaping the CRC immune microenvironment.

## Relationship between LNM-related variations in gut microbiota and immune-associated genes

By analyzing the correlation between LNM-associated differential gut microbiota and common immune-related genes, we discovered a potential link between LNM-associated gut microbiota and host immunity. As shown in Fig. 5, in NLNM-associated bacteria, g__Clostridium_XVIII.s__Clostridium_ramosum showed significant negative correlations with multiple immune checkpoints (CD27, BTLA) (Fig. 5A), chemokines (CXCL6, CXCL5) (Fig. 5B), immune activation genes (TNFRSF17, IL2RA) (Fig. S1A), immune suppressor genes (KIR2DL1, BTLA) (Fig. S1B), and chemokine receptors (CXCR1, CXCR2) (Fig. S1C). In LNM-associated bacteria, g__Bacteroides.s__Bacteroides_plebeius exhibited significant positive correlations with various immune checkpoints (TNFSF9, HHLA2) (Fig. 5C), chemokines (CXCL8, CCL7) (Fig. 5D), immune activation genes (TNFSF9, HHLA2) (Fig. S1D), immune suppressor genes (IL10RB) (Fig. S1E), and chemokine receptors (CXCR1, CXCR2) (Fig. S1F). Given the crucial role of the human immune system in tumor development, these findings suggest that LNM-associated differential gut microbiota may influence the expression of immune-related genes.

## Identification of LNM-associated biological functional pathways and correlation of differential pathways with differential gut microbiota

In order to identify the differential biological functional pathways between LNM and NLNM groups and to investigate the correlation between different gut microbiota in two groups and these pathways, we utilized single sample gene set enrichment analysis (ssGSEA) to convert the gene expression matrix obtained from transcriptome sequencing of tumor tissue samples from 23 CRC patients and the species abundance matrix obtained from 16S rRNA sequencing of gut microbiota into corresponding score matrices. Subsequently, these matrices were analyzed through KEGG and Gene Ontology (GO) analyses, with GO analysis including cellular component (CC), molecular function (MF), and biological process (BP). After performing differential analysis on the GO (Fig. 6A) and KEGG (Fig. 6B) pathway score matrices for the LNM and NLNM groups, in the LNM group, we identified 65 significantly upregulated GO pathways [such as GOBP: leukotriene biosynthetic process (logFC = 0.038, $P = 0.0009$)] and two significantly upregulated KEGG pathways [such as KEGG: arachidonic acid metabolism (logFC = 0.022, $P = 0.017$)]. In the NLNM group, a total of 554 significantly upregulated GO pathways were identified [such as GOBP: establishment of endothelial intestinal barrier (logFC = 0.048, $P = 0.0001$)] and 16 significantly upregulated KEGG pathways [such as KEGG: nicotinate and nicotinamide metabolism (logFC = 0.028, $P = 0.002$)]. The complete information of the GO and KEGG enrichment items can be found in Table S3. These findings highlight the different biological functions between LNM and NLNM. We further investigated the correlation coefficients between each group of bacteria and BP items, MF items, and KEGG pathways (Table S4) and found that there is a significant correlation between certain gut microbiota and biological pathways (Fig. 6C). For example, in the

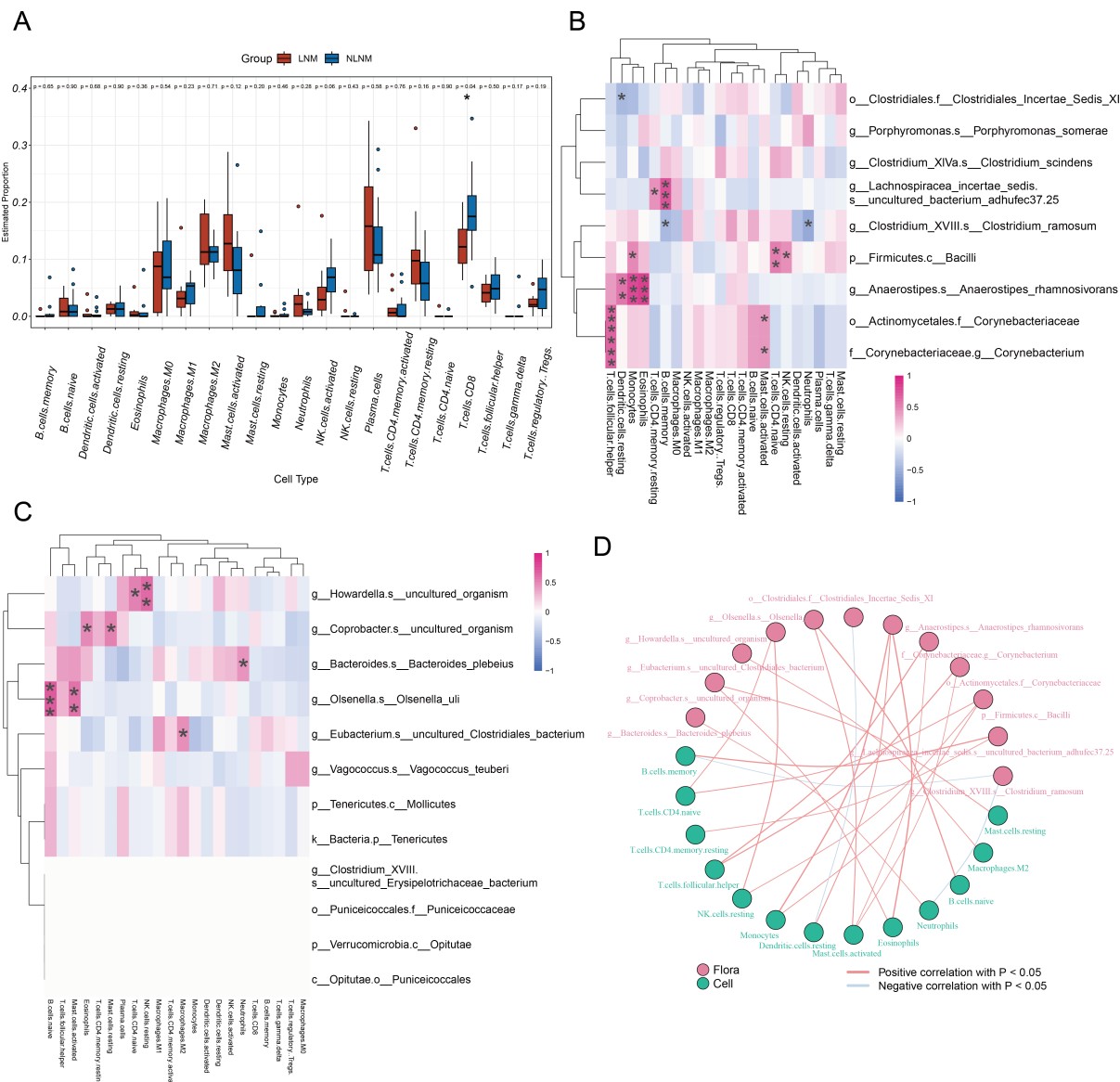

**FIG 4** Relationship between LNM-associated gut microbiota and infiltrating immune cells in CRC. (A) Box plot of estimated proportion of immune cells in CRC patients grouped by LNM and NLNM. Horizontal coordinates represent immune cells in 22, and vertical coordinates are their estimated proportion. Red color indicates the LNM group, and blue color indicates the NLNM group. In the graph, "*" indicates a *P*-value <0. 05 and "**" indicates a *P*-value <0. 01. (B) Heat maps of correlation between dominant colonies in NLNM and tumor-infiltrating immune cells. (C) Heat maps of correlation between dominant colonies in LNM and tumor-infiltrating immune cells. The horizontal coordinates are immune cells, and the vertical coordinates are bacteria. Red indicates a positive correlation, and blue indicates a negative correlation. The depth of the color indicates the size of the Pearson correlation coefficient, and the color from light to dark indicates the value of the phase relationship from small to large. The "*" in the graph represents the size of the *P*-value: no * for *P*-value ≥0.05, * for 0.01 ≤ *P* < .05, ** for 0.001 ≤ *P* < .01, *** for *P* < 0.001. (D) Network diagram of correlations between gut microbiota and immune cell differences associated with LNM status. Nodes in different colors in the figure represent gut microbiota and immune cells, respectively, and the connection lines between nodes indicate significant correlation between nodes. The blue line indicates that the Spearman correlation coefficient is less than 0 (negative correlation), while the red line indicates that the Spearman correlation coefficient is greater than 0 (positive correlation).

LNM group, *g__Bacteroides.s__Bacteroides_plebeius* has a statistically significant positive correlation with GOBP: inosine monophosphate (IMP) metabolic process (*r* = 0.59, *P* = 0.003). In the NLNM group, *g__Porphyromonas.s__Porphyromonas_somerae* has a statistically significant positive correlation with GOBP: regulation of telomere capping (*r* = 0.62, *P* = 0.001). These results suggest that the differential gut microbiota associated

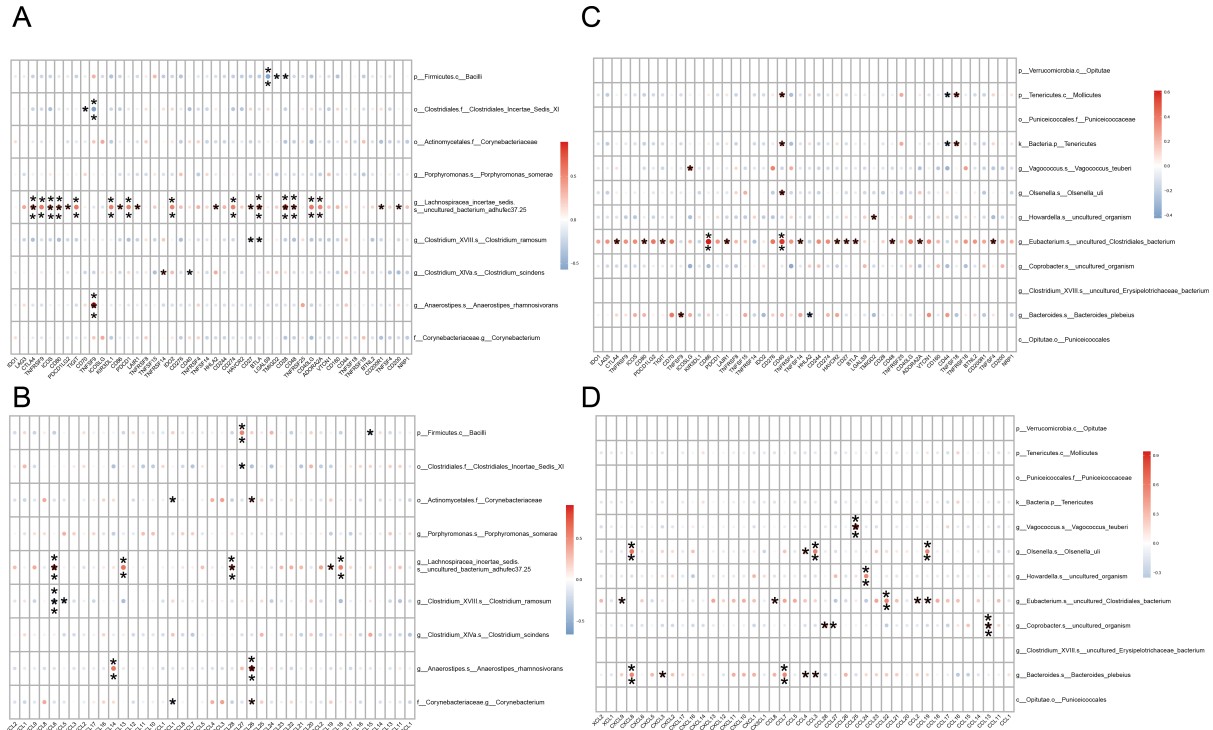

**FIG 5** Correlation between LNM-related differences in gut microbiota and immune-related genes. (A) Heat map of correlation between the dominant microbiota and immune checkpoints in NLNM. (B) Heat map of the correlation between the dominant microbiota and chemokines in NLNM. (C) Heat map of correlation between the dominant microbiota and immune checkpoints in LNM. (D) Heat map of the correlation between the dominant microbiota and chemokines in LNM. In the figure, the horizontal coordinate is the immune-related genes, and the vertical coordinate is the bacteria. In the figure, the red indicates the positive correlation, and the blue indicates the negative correlation. The color depth indicates the size of the Pearson correlation coefficient, and the color from light to dark indicates the value of the phase relationship from small to large. The "*" in the graph represents the size of the *P*-value: no * for *P*-value ≥0.05, * for 0.01 ≤ *P* < .05, ** for 0.001 ≤ *P* < .01, *** for *P* < 0.001.

with LNM may influence the metastasis of cancer cells to lymph nodes by participating in certain potential biological functional pathways.

## Construction of a model to predict LNM status using differential gut microbiota characterization

As shown in Fig. 2B, LEfSe analysis identified 21 differential gut microbiota associated with LNM status. After ranking the importance of gut microbiota features (Fig. S2), all 21 were retained to develop RF and MLP models based on LNM-associated gut microbiota. One hundred forty-seven CRC patients' gut microbiota samples with LNM labels were randomly split into training and validation sets at a 7:3 ratio.

In the RF-based LNM prediction model, the training set confusion matrix (Fig. 7A) revealed a predominance of true-negative (TN) and true-positive (TP) cases over false-negative (FN) and false-positive (FP) classifications, indicating robust classification accuracy (area under the curve [AUC] = 0.957) and stability in distinguishing LNM-positive and -negative samples. In the validation set (Fig. 7B), despite a performance decline, TN and TP cases remained higher than FN/FP, with an AUC of 0.705 (Fig. 7C), suggesting retained generalizability for clinical application.

For the MLP-based model, similar trends were observed in the MLP training set confusion matrix (Fig. 7D), with TN/TP cases significantly outnumbering FN/FP (AUC = 0.947). In the validation set (Fig. 7E), performance markedly decreased (AUC = 0.625) (Fig. 7F), indicating the MLP model may be less generalizable and stable than the RF model, with performance being more dependent on data quality and feature selection.

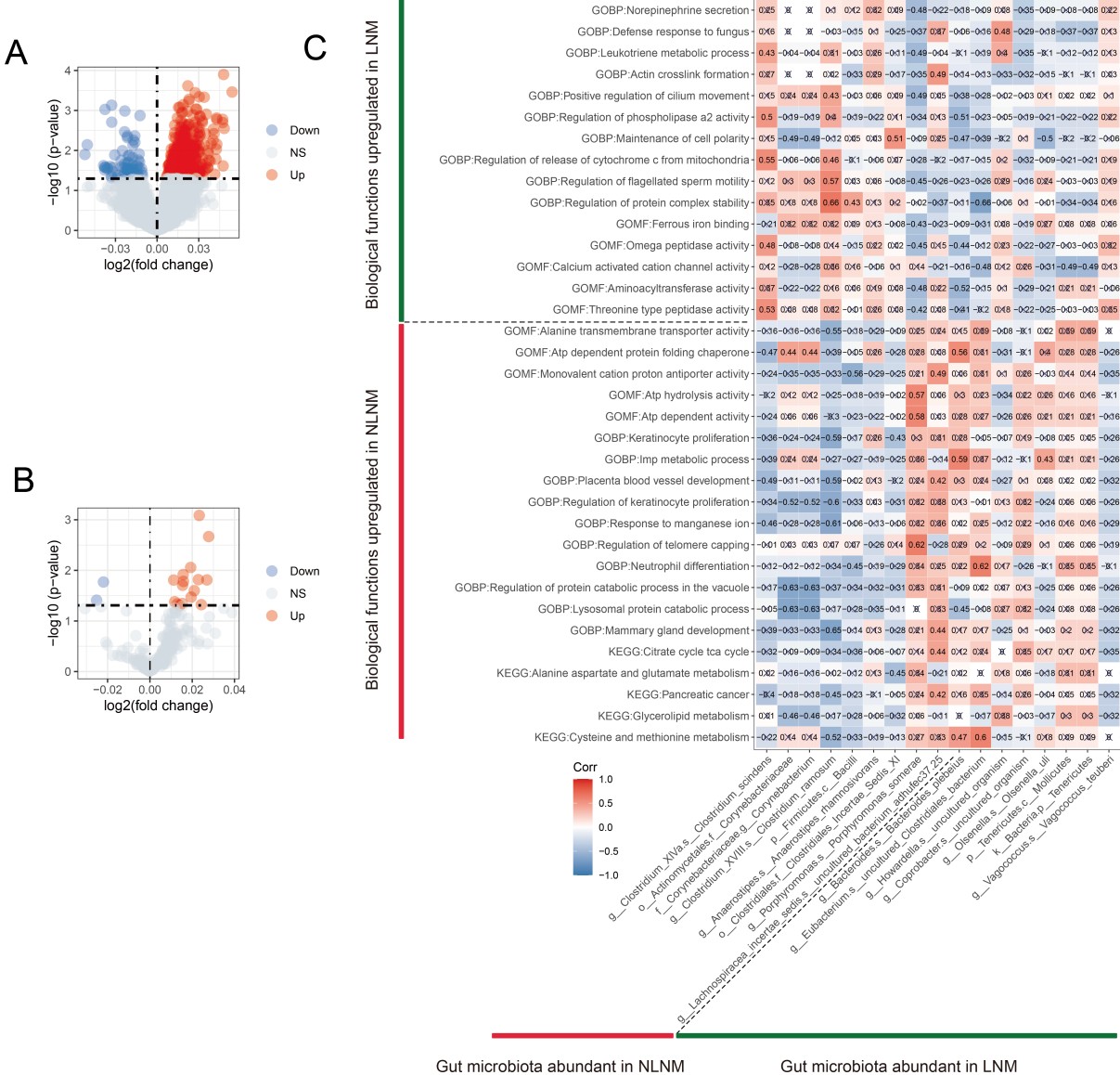

**FIG 6** Identification of LNM-related differential pathways and their correlation with LNM-related differential gut microbiota. (A) GO volcano plot of the associated differential expression in NLNM versus LNM. (B) KEGG volcano plot of the associated differential expression in NLNM versus LNM. The horizontal coordinates indicate log 2 (fold change). The farther the point is from the center, the larger the multiple of the difference; the vertical coordinates represent -log 10 (P-value). The closer the point is to the top, the more significant the difference in expression. Each dot represents the differentially expressed gene detected. Red indicates upregulated genes. Blue indicates downregulated genes. Gray indicates no differential genes. (C) LNM-related differences correlation graph of gut microbiota with LNM-related differences in BP and MF. The horizontal coordinate of the graph is bacteria. The vertical coordinates are the GOBP and GOMF items. In this figure, red indicates positive correlation, blue indicates negative correlation, color depth and numerical value indicate the size of the Spearman correlation coefficient, and color from light to dark indicates the value of phase relationship from small to large. The symbol "×" in the figure represents the size of the P-value: the presence of "×" means P-value ≥0.05, and the absence of "×" means P-value <0.05.

In summary, both models showed some effectiveness in predicting LNM in patients, with the RF model performing better in balancing training and validation performance. The MLP model, while good on the training set, had a more pronounced drop in validation set performance.

## DISCUSSION

Having employed 16S rRNA sequencing and transcriptome sequencing technologies, we meticulously analyzed the community diversity of gut microbiota and differences in dominant bacterial groups between LNM and NLNM CRC patients. We clarified the relationship between dominant bacterial groups and host immunity and deeply explored the potential biological pathways associated with LNM. After identifying differential gut microbiota, we then used these microbial features to construct a classification model to predict LNM status.

The β-diversity analysis revealed two distinct clusters separating LNM and NLNM groups, demonstrating significant divergence in gut microbial composition. Previous studies have noted that *Fn* is enriched in the tumor tissue of LNM-group CRC patients (15, 16) and that the microbiota plays a role in CRC-LNM (18), but they did not explore how the microbiota affects LNM. Our study further explored potential key fecal microbes driving LNM and assessed their feasibility and efficacy as biomarkers.

At the species level, we found that *Bacteroides plebeius* was enriched in the gut microbiota of LNM. *B. plebeius*, belonging to the *Bacteroidetes* phylum, is a bacterium successfully isolated from the human gut that consumes seaweed (19). Compared with the gut microbiota of healthy adults (20), the abundance of *B. plebeius* is significantly increased in the gut of CRC patients and those with adenomatous polyps. Further research investigating the composition of the gut microbiota during the transformation of colorectal adenomas to cancer revealed the importance of 10 key species, including *B. plebeius* (21). In the analysis of the gut microbiota of Lynch syndrome patients

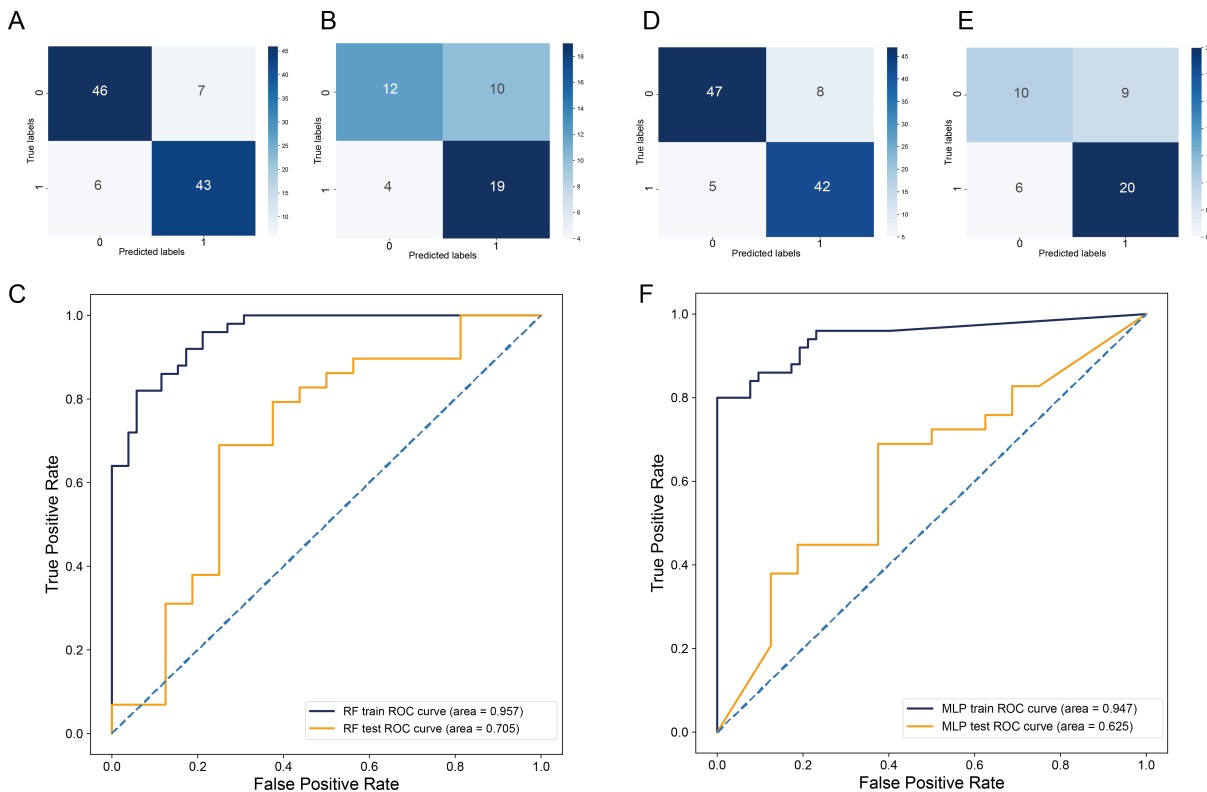

**FIG 7** Machine learning models based on LNM-associated differential gut microbiota to predict LNM status in CRC. (A) Confusion matrix in the training set of the RF model. (B) Confusion matrix in the RF model testing set. (D) Confusion matrix in the training set of the MLP model. (E) Confusion matrix in the MLP model testing set. The horizontal axis represents the model's prediction label, while the vertical axis represents the true sample status. The values "1" and "0" signify positive and negative predictions, respectively. The numbers within different boxes represent the sample count. The color depth is proportional to the number of samples; the greater the number of samples, the darker the color depth. (C) Receiver operating characteristic (ROC) curves of the RF model training set and testing set. (F) ROC curves of the MLP model training set and testing set. The AUC values in the figures represent AUC values, with higher AUC values indicating better model prediction performance.

prone to colorectal cancer (22), *B. plebeius* showed overproliferation in the gut. These findings suggest that this bacterium may play a promotional role in the occurrence and development of colorectal cancer. In seaweed-treated mice, *B. plebeius* binds to IgA, which can limit bacterial growth through the chaining and aggregation of dividing cells (23), a phenomenon consistent with our study. Although it has been speculated that it may accelerate the progression of solid tumors by suppressing the body's anti-tumor immune response (24), to date, there have been no in-depth studies directly targeting the specific mechanisms between *B. plebeius* and the development of CRC. Overall, as a relatively novel bacterium in CRC research, *B. plebeius* is still in the early stages of study and requires further exploration. It is worth noting that in our predictive model construction process, *B. plebeius* ranked first in the gut microbiota feature ranking, and previous studies as well as our own research have shown that it is significantly enriched in the gut of CRC patients. This finding suggests that *B. plebeius* has the potential to become an important biomarker for identifying whether CRC patients have LNM.

In order to identify the enriched biological pathways associated with the gut microbiome in relation to LNM status in CRC, we utilized the PICRUSt2 software to predict the differential KEGG pathways between the gut microbiomes of NLNM and LNM-CRC patients. Our findings revealed that endocytosis is significantly enriched in the NLNM group. In immune defense, macrophages can precisely use receptor-triggered endocytic pathways (25) to internalize and degrade bacterial pathogens while presenting their antigens to stimulate the body's immune response. Clathrin-mediated endocytosis and clathrin-independent endocytosis critically regulate T cell functionality by facilitating the internalization and recycling of T cell antigen receptors (TCRs) (26), a process indispensable for the activation of naïve CD4+ and CD8+ T cells. Notably, the intracellular pool of CD3ζ—a key component of TCR signaling—relies on endocytic trafficking for its formation and maintenance (27). Dysregulation of this process compromises immune competence, as evidenced by impaired T cell responses in endocytosis-deficient models. Moreover, in interactions with the microbiota, endocytosis facilitates the entry of beneficial molecules released by symbiotic bacteria, such as *Bifidobacterium*, into intestinal epithelial cells (28), exerting anti-inflammatory (29) and antioxidant effects, strengthening the gut barrier, and maintaining microbial equilibrium, which is significant for CRC prevention. Novel strategies for delivering targeted drugs via endocytic pathways have been discovered, such as using vitamin B12 overexpressed in colorectal cancer cells to design conjugates that precisely inhibit tumor growth (30), demonstrating the potential therapeutic applications of endocytosis. Overall, the upregulation of endocytosis in NLNM-CRC patients may enhance their anti-tumor immune capacity, potentially explaining the limited tumor progression and reduced metastasis in these patients. Although our functional predictive analysis highlighted differences in microbial pathway abundance, we acknowledge that the study did not explicitly assess prevalence bias and was limited by the lack of prevalence analysis for different subgroups such as age, race, or geographic region. In the future, multicenter studies on larger stratified populations are needed to validate these findings and evaluate their applicability to a wider population.

The immune response features of the tumor microenvironment greatly influence CRC progression and treatment response (31). In terms of research methods, future research has adopted clustering validation methods such as silhouette analysis to evaluate the robustness of identified immune cell subpopulations. In this study, we found that CD8+ T cells infiltration was much higher in the NLNM group than in the LNM group, consistent with previous studies linking high CD8+ T cell abundance to longer overall survival (OS) and disease-free survival (DFS) in CRC patients (32). This may be because CD8+ T cells boost PD-1/PD-L1 expression (33), enhancing sensitivity to immune checkpoint inhibitors. Fecal microbiota is associated with CRC T cell infiltration (18), with specific gut bacteria like the *Faecalibacterium* genus promoting CD8+ T cell differentiation into effector memory phenotypes through metabolites (34), thus strengthening anti-tumor immunity. This indicates complex immune-microbiota

interactions in cancer. For instance, in our study, *B. plebeius* was positively correlated with neutrophils. Neutrophils drive cancer progression through multiple pathways. Chemokines like CXCL-8 and CXCL1/2/5 in the tumor microenvironment attract neutrophils to infiltrate late-stage tumors, where they release reactive oxygen species (ROS) to induce DNA damage and secrete vascular endothelial growth factor (VEGF) to accelerate angiogenesis (35, 36). Neutrophil-derived arginase 1 depletes microenvironmental arginine, inhibiting the CD3ζ signaling pathway and blocking CD8+ T cell activation (37), which aids cancer growth. Neutrophils with high β2-integrin expression promote CRC liver metastasis by delivering pro-metastatic signals via extracellular vesicles (38), and the signaling niches they form also boost cancer invasiveness (39). CRC cells may attract neutrophils due to the high infiltration of *Fusobacterium* and *Bacteroides* genus in the microenvironment (40), which aligns with our findings. In summary, CRC progression likely involves complex regulatory networks of microbiota-immune interactions.

In terms of microbiota-immunity gene correlation, *B. plebeius* became significantly positively correlated with CXCL-8. CXCL8, also known as IL-8, is an important chemokine whose role in CRC distant metastasis has been extensively studied. Angiogenesis is key to CRC progression, and CXCL8 promotes CRC angiogenesis through the CXCR2 receptor, providing survival resources (41) for the tumor and creating conditions for metastasis. CXCL8 can also act in concert with CCL20 to synergistically induce epithelial-mesenchymal transition (EMT) through the PI3K/AKT-ERK1/2 signaling axis, thereby promoting CRC metastasis progression (42). In addition, studies have found that *Streptococcus gallolyticus* can stimulate colon cancer cells to release IL-8 (43), while *Fusobacterium nucleatum* induces the secretion of IL-8 and CXCL1 by host cell binding and invasion (44), promoting the migration of colorectal cancer cells and highlighting the synergistic role of the gut microbiota and chemokines in CRC. The upregulation of IL-8 in CRC epithelial cells has also been confirmed (45). Combined with our correlation analysis, it can be speculated that the potential interaction between *B. plebeius* and IL-8 may be one of the possible reasons for CRC to undergo lymphatic metastasis and further distant metastasis.

In the analysis of the differential biological pathways between two groups of patients, we also found that *B. plebeius* is significantly positively correlated with the GOBP: IMP metabolic process ($r = 0.59$, $P = 0.003$). This suggests that *B. plebeius* may play a pivotal role in modulating nucleotide biosynthesis during CRC progression. As a central hub in purine metabolism, IMP serves as a precursor for AMP and GMP synthesis, fueling rapid DNA/RNA production essential for tumor cell proliferation and metastasis. Notably, inosine 5′-monophosphate dehydrogenase type II (IMPDH2), the rate-limiting enzyme in IMP-to-GMP conversion, is overexpressed in multiple cancers and correlates with poor prognosis (46, 47). Mechanistically, IMPDH2 facilitates G1/S phase transition via PI3K/AKT/mTOR and FOXO1 signaling while concurrently promoting EMT and chemoresistance (48). In CRC, pharmacological inhibition of IMPDH2 suppresses tumor growth (49) and reverses oxaliplatin resistance (50), underscoring its therapeutic potential. However, causality remains undetermined as to whether *B. plebeius* directly regulates IMP metabolism or simply grows in IMP-rich niches and needs to be verified by abiotic mouse models and *in vitro* co-culture systems.

LNM is a core indicator in CRC clinical staging, determining treatment strategies and prognosis assessment for stage III patients. However, preoperative LNM diagnosis mainly relies on imaging examinations with low sensitivity and lacks precise prediction tools for micrometastasis. Recent studies have focused on developing prediction models based on genomics or radiomics but overlooked the gut microbiota, a dynamic factor regulating the tumor microenvironment. In this study, we integrated LNM-associated microbial features with machine-learning algorithms to build non-invasive prediction models based on RF and MLP. The models showed excellent performance in the training set (AUC > 0.90) and the validation set, with better stability for the RF model (AUC = 0.705 vs MLP = 0.625). This difference may be due to RF's robustness in handling high-dimensional nonlinear feature interactions. Importantly, our model requires only

a stool sample for non-invasive detection, offering a low-cost, easy-to-implement, and efficient strategy for predicting LNM in CRC.

The gut microbiota has emerged as a pivotal target in CRC precision oncology, offering dual utility as a predictive biomarker and a therapeutic modulator. For instance, *Bifidobacterium breve* abundance predicts clinical response to PD-1 inhibitor-chemotherapy combinations in advanced non-small cell lung cancer (51), while FMT restores microbial homeostasis and enhances anti-tumor immunity in CRC patients (52). Our study further identifies LNM-associated microbial signatures (e.g., *Bacteroides plebeius*) that could guide microbiota-targeted therapies, such as probiotic engineering or precision FMT donor selection.

However, this study also has certain limitations. First, the study focused only on the comparative analysis of fecal microbiota in patients with LNM-positive versus -negative CRC and lacked data from healthy controls as well as systematic comparisons between mucosa-associated microbiota and fecal microbiota. Meanwhile, while our study employed multivariate LEfSe analysis to adjust for known confounders (e.g., age, gender, and body mass index), residual confounding from unmeasured variables such as dietary habits, medication use, or gut microenvironment heterogeneity may persist. These factors could influence both gut microbiota composition and tumor progression, potentially biasing our results. Secondly, intratumoral microbiota has been increasingly studied in recent years and has been shown to be closely associated with the development of several cancers (53) and to play a potential role in modulating tumor sensitivity to chemotherapeutic agents (54), whereas the analysis of intratumoral microbiota was not addressed in this study. In view of this, the present study has certain deficiencies in the breadth and depth of microbial community analysis. In addition, this study was mainly based on correlation analysis, and the conclusions obtained still remained at the hypothesis level, which urgently needs to be verified by further wet experiments. In the future, we plan to incorporate samples from CRC lesions and adjacent normal mucosa into our study, as well as to analyze the composition and function of the microbial community in conjunction with the detection of the bacterial flora in tumor tissues, and to validate the relevant mechanisms through animal models or cellular experiments, so as to reveal the similarities and differences in the microbiology of diseased tissues and normal tissues with greater precision.

## Conclusion

In CRC patients, we identified 21 gut microbiota significantly associated with LNM, particularly *Bacteroides plebeius*, which was significantly enriched in the LNM group and closely related to LNM-CRC. This species may play a key role in accelerating the progression of CRC and promoting lymphatic and distant metastasis. This process may involve an increase in neutrophils and the upregulation of the expression of CXCL8 and IMP metabolic pathways. Furthermore, based on the characteristics of LNM-associated gut microbiota, the RF and MLP models we constructed showed significant clinical application prospects in predicting the LNM status of CRC patients.

## MATERIALS AND METHODS

### Subject information and sample collection

We successfully collected fecal samples from 236 CRC patients treated at the hospital from 1 January 2021 to 31 December 2021. After strict screening, 198 samples met the quality requirements for 16S rRNA sequencing. Subsequently, we meticulously extracted key demographic and clinical pathological characteristics from the medical records of these patients, including gender, age, TNM staging, tumor size, and microsatellite stability status. Specifically, in this study, we conducted an in-depth analysis of 147 fecal samples carrying information on lymph node metastasis status, specifically distinguish-

ing 69 NLNM cases from 78 LNM cases. In addition, we carefully collected postoperative tumor samples from 23 participants for further transcriptome sequencing analysis.

The strict criteria for sample inclusion ensured the quality and reliability of the research data, specifically including (i) patients who underwent surgery and received a clear pathological staging according to the 8th edition of the AJCC CRC staging criteria or those diagnosed with colorectal adenocarcinoma through colonoscopic biopsy; (ii) no history of other malignant tumors; (iii) no history of intestinal diseases or acute complications such as intestinal perforation, intestinal obstruction, pelvic abscess, etc.; (iv) no anti-tumor treatment of any kind received prior to fecal sample collection; (v) no use of antibiotics or gut microbiota modulators within 1 month prior to sample collection; (vi) no history of consciousness disorders or other cognitive function impairments.

The fecal sample collection was carried out on the 1st day of the patient's admission. We specifically instructed patients to collect the middle section of the stool using a sterile collection tube to prevent the mixture of urine. After collection, the samples were quickly aliquoted into 2 mL Eppendorf (EP) tubes (each containing approximately 200 mg) and immediately placed in a −80°C sterile ice box for long-term freezing preservation. At the same time, the freshly resected tumor tissue (about the size of a soybean, with a diameter of 3–5 mm) was properly stored in liquid nitrogen within 30 minutes of acquisition to maintain its biological activity and integrity.

## 16S rRNA sequencing and gut microbiota analysis

Before formal sequencing, we first conducted a thorough assessment of the DNA quality in the fecal samples. Using the MOBIO PowerSoil DNA Extraction Kit, 200 mg of fecal samples were mixed with Tris-EDTA buffer to optimize the DNA extraction process. Subsequently, only high-quality DNA samples were selected for PCR amplification to ensure the accuracy of subsequent analyses. During the amplification process, we used specific primers 341F (5′-CCTACGGGNGGCWGCAG-3′) and 805R (5′-GACTACHVGGGTATC-TAATCC-3′), which precisely target the V3 and V4 regions of the 16S rRNA gene, achieving selective amplification of the target fragments (55). The amplified products were then identified by 2% agarose gel electrophoresis, focusing on bands in the range of 300 to 350 bp to ensure the accuracy and specificity of the amplified fragments. Using the Quant-iT PicoGreen dsDNA assay kit, we accurately determined the concentration of the PCR products, and all samples were mixed at equimolar concentrations. Subsequently, the combined samples were quantified a second time using the KAPA Library Quantification Kit KK4824 to ensure consistency of the samples before sequencing.

On the Illumina PE250 platform, we conducted high-throughput sequencing of qualified libraries using $2 \times 250$ bp chemistry reagents. After obtaining raw sequencing data in FASTQ format, we used QIIME2 for quality control, noise reduction, species annotation, and filtering of low-abundance and contaminant sequences. This process removed low-quality sequences, retaining high-quality data for subsequent analyses.

We annotated the gut microbiota using the Greengene database v.13.8 and performed amplicon sequence variant (ASV)/operational taxonomic unit (OTU) extraction and analysis in the phyloseq package v.1.26.1. The α-diversity indices (Chao1, ACE, Shannon, and Simpson) were used to describe microbial richness and evenness within individual samples, while β-diversity analysis compared microbial community structures between different samples, revealing their similarities and differences. Bray-Curtis dissimilarity and Jaccard index were used to quantify differences in species composition between samples, with results analyzed using ADONIS and ANOSIM in the vegan package v.2.5.6.

To further explore the classification and comparative information within high-dimensional data, we utilized the LEfSe software v.1.0.0 to screen out gut microbial communities that significantly differentiate between lymph node metastasis and non-lymph node metastasis CRC patients under a strict threshold of LDA score |LDA score| > 2 and $P$-value <0.05. The influence of these differentially abundant microbial communities was

assessed through LDA scores and, after logarithmic transformation, presented in an intuitive bar chart format using the ggplot2 package v.3.4.0.

For the identified key gut microbial communities, we further utilized the PICRUSt2 software v.2.3.0 to predict the potential enrichment of KEGG pathways between LNM-CRC and NLNM-CRC based on 16S rRNA sequencing data. By employing the non-parametric Mann-Whitney U-test, we compared the differences in α-diversity indices between the two groups and, with the aid of the vegan package v.2.5.6, conducted an in-depth analysis of the enrichment of KEGG pathways between the groups. All data analyses were performed in the R software v.4.3.2 environment, with statistical significance set at a *P*-value <0.05 (two-tailed test).

## Transcriptome sequencing

RNA was extracted from tissue samples of 23 CRC tumors using the Trizol Total RNA Extraction Kit, 11 of which were from patients with lymph node metastasis and 12 from patients without lymph node metastasis. The integrity of the isolated RNA was verified by electrophoresis, and the purity was assessed using a mini UV-spectrophotometer. After removing rRNA, cDNA libraries were prepared according to the guidelines of the RNA-seq Sample Preparation Kit (VAHTS Stranded mRNA-seq Library Prep Kit for Illumina). These transcriptome libraries were sequenced on the Illumina NovaSeq 6000 system, generating 6G of data per sample. The quality of the sequencing data was evaluated using FastQC. Subsequently, the obtained sequences were aligned with the reference genome (hg38 version) using HISAT2 for gene expression analysis. StringTie was used to quantify the gene expression levels based on established gene models, and the expression abundance of each gene was calculated in terms of transcripts per million (TPM).

## Analysis of the CRC tumor immune microenvironment

CIBERSORT is an algorithm for quantifying the composition of immune cells from RNA sequencing data, relying on the unique gene expression signatures of different immune cells and employing a machine-learning algorithm to identify and classify these gene expression profiles (56). We used the CIBERSORT R version, in conjunction with known reference gene expression profiles and gene expression data from the composite samples to be analyzed, to create a model using support vector regression. The TPM matrix derived from the transcriptome sequencing was transformed into a matrix representing the relative proportions and functional states of 22 different immune cell types. The microbial composition matrix and the immune cell proportion matrix were integrated, and the correlation coefficients between each column of the merged matrix were calculated using the rcorr function in R.

## Functional enrichment analysis of LNM-related transcriptome sequencing

In the R software v.4.3.2 environment, we first conducted an in-depth analysis of RNA sequencing data by applying the ssGSEA algorithm to gene sets in GMT format (downloaded from the GSEA website: https://www.gsea-msigdb.org/gsea/index.jsp, including c2.cp.kegg.v2022.1.Hs.symbols.gmt and c5.go.v2022.1.Hs.symbols.gmt). This algorithm calculates ssGSEA scores for each gene set in a sample based on the descending order of gene expression levels, quantifying the extent of coordinated upregulation or downregulation of members of a particular gene set in the sample. Subsequently, we utilized the "ssgsea" function in the "GSVA" package v.1.46.0 to generate a gene set score matrix for each sample. To identify differences between lymph node metastasis and non-lymph node metastasis, we set the non-lymph node metastasis group as the control group and employed the "limma" algorithm integrated in the "TCGAbiolinks" package v.2.25.3 to perform a differential analysis of GO items and KEGG pathways between the two groups, with the selection criteria set to *P*-value <0.05. The GO analysis covered three aspects: BP, MF, and CC. To visually present these significantly different GO terms and

KEGG pathways, we used the powerful plotting capabilities of the "ggplot2" package v.3.4.0 to create volcano plots, effectively visualizing the analysis results.

## Machine learning model building and identification of gut microbiota markers

We designed and implemented two machine learning architectures, RF and MLP, aimed at predicting LNM status of CRC patients based on gut microbiota features. RF is an ensemble machine learning method based on decision trees (57), which improves the overall model's predictive accuracy and stability by constructing multiple decision trees and combining their predictions. Its inherent feature selection mechanism is particularly powerful, and there have been studies exploring strategies to further optimize RF performance by reducing the feature space, and these efforts have successfully improved the model's predictive accuracy in some scenarios (58). RF not only occupies an important position in the modern field of machine learning but is also considered one of the cutting-edge technologies for building predictive models, showing a significant advantage in predictive performance compared to traditional methods (59–61). MLP is a neural network-based machine learning model that performs high-level abstraction and classification of input data through multiple layers of nonlinear transformations. MLP typically consists of an input layer, one or more hidden layers, and an output layer, each containing multiple neurons that are interconnected through weights and biases (62). MLP is not only short in training time and easy to implement but also provides a more robust model (63), and it can solve more problems than a single-layer perceptron model. Therefore, MLP has shown unique advantages and potential in solving complex prediction tasks.

Using Python in conjunction with the SciKit Learn 1.0.2 library (https://scikit-learn.org/stable/), we set up a machine learning modeling environment and configured the model parameters by default. Based on 147 CRC patient gut microbiota samples with LNM labels, we randomly split the data set into training and validation sets at a 7:3 ratio. Using 21 differential gut bacteria as features, we constructed RF and MLP models. We evaluated model performance on both sets using ROC curves and AUC values to ensure comprehensive and accurate assessment.

## Statistical methods

In R software version 4.3.2, we conducted statistical assessments using the Pearson $\chi^2$ test for categorical data and $t$-tests for continuous data related to clinical baseline characteristics. To establish correlations between the gut microbiome, immune cell counts, and immune-related gene expression, we performed Pearson correlation analysis using the Hmisc package version 5.1.1. The ggcorrplot software package version 0.1.4.1 was utilized for Spearman correlation analysis to examine the relationships between the dominant microbial groups and the enriched BPs, MFs, and KEGG pathways. We employed various tools for visualizing the analysis results: pheatmap version 1.0.12 for heat maps, ggcorrplot package version 0.1.4.1 for correlation plots, ggplot2 version 3.4.0 for volcano plots, Igraph package version 1.6.0 for network graphs, and Cytoscape software version 3.10.1 for complex network analysis.

## ACKNOWLEDGMENTS

This work was supported by the Natural Science Foundation of Guangxi Province (Guangxi Natural Science Foundation) (2021GXNSFAA196008), Youth Science Foundation of Guangxi Medical University (GXMUYSF202402), Youth Research Project of Guangxi Medical University Affiliated Cancer Hospital (yuanqingji2023-10hao), Middle-aged and Young Teachers' Basic Ability Promotion Project of Guangxi (Basic Ability Promotion Project of Guangxi) (2021KY0087), China Postdoctoral Science Foundation (2023MD734155), and Youth Science Foundation of Guangxi Medical University (GXMUYSF202357).

Yongzhi Wu, Chengen Deng, Xiaoliang Huang, Zhen Wang, Zigui Huang, Jungang Liu, Weizhong Tang: conceived and designed the experiments; Jungang Liu, Xiaoliang Huang, Yongzhi Wu, Yongqi Huang, Zigui Huang, Fuhai He, Chuanbin Chen, Mingjian Qin, Chenyan Long, Weizhong Tang: analyzed the data; Jungang Liu, Xiaoliang Huang, Yongzhi Wu, Yongqi Huang, Zigui Huang, Zhen Wang, Mingjian Qin, Fuhai He, Chuanbin Chen, Shenghai Liu, Rumao Zhong, Jun Liu, Chenyan Long, Weizhong Tang: helped with reagents/materials/analysis tools; Jungang Liu, Xiaoliang Huang, Yongzhi Wu, Yongqi Huang, Chuanbin Chen, Zigui Huang, Mingjian Qin, Chenyan Long, Weizhong Tang: contributed to the writing of the manuscript. All authors reviewed the manuscript.

## AUTHOR AFFILIATIONS

[1]Division of Colorectal & Anal Surgery, Department of Gastrointestinal Surgery, Guangxi Medical University Cancer Hospital, Nanning, Guangxi, People's Republic of China
[2]Department of Urology, Guangxi Medical University Cancer Hospital, Nanning, Guangxi, People's Republic of China

## AUTHOR ORCIDs

Yongzhi Wu http://orcid.org/0009-0009-3224-8258
Zhen Wang http://orcid.org/0000-0002-8282-9781
Jungang Liu http://orcid.org/0000-0002-1602-6235
Weizhong Tang http://orcid.org/0000-0003-0877-8557
Xiaoliang Huang http://orcid.org/0000-0001-7979-7398

## FUNDING

| Funder | Grant(s) | Author(s) |
| --- | --- | --- |
| Youth Science Foundation of Guangxi Medical University | GXMUYSF202402 | Xiaoliang Huang |
| Natural Science Foundation of Guangxi Zhuang Autonomous Region | 2021GXNSFAA196008 | Jungang Liu |
| Middle-aged and Young Teachers' Basic Ability Promotion Project of Guangxi | 2021KY0087 | Jungang Liu |
| China Postdoctoral Science Foundation | 2023MD734155 | Jungang Liu |
| Youth Science Foundation of Guangxi Medical University | GXMUYSF202357 | Chenyan Long |
| Guangxi Zhuang Autonomous Region Medical Young Reserve Talents Training Program | | |

## AUTHOR CONTRIBUTIONS

Yongzhi Wu, Conceptualization, Data curation, Formal analysis, Investigation, Resources, Software, Supervision, Validation, Writing – original draft, Writing – review and editing | Chengen Deng, Resources, Validation, Writing – original draft, Writing – review and editing | Zigui Huang, Investigation, Resources, Software, Supervision, Writing – original draft, Writing – review and editing | Yongqi Huang, Data curation, Resources, Software, Writing – review and editing | Chuanbin Chen, Writing – review and editing | Mingjian Qin, Writing – review and editing | Zhen Wang, Writing – review and editing | Fuhai He, Writing – review and editing | Shenghai Liu, Data curation, Formal analysis, Writing – review and editing | Rumao Zhong, Writing – review and editing | Jun Liu, Writing – review and editing | Chenyan Long, Writing – review and editing | Jungang Liu, Writing – review and editing | Weizhong Tang, Writing – review and editing | Xiaoliang Huang, Conceptualization, Funding acquisition, Investigation, Writing – original draft, Writing – review and editing

## DATA AVAILABILITY

The original contributions presented in the study are included in the article material. The 16S rRNA data that support the findings of this study are openly available in the National Genomics Data Center (NGDC) database under accession number HRA011714. Transcriptome data are publicly available at under accession number HRA011782.

## ETHICS APPROVAL

This study was approved by the Ethics and Human Subject Committee of Guangxi Medical University Cancer Hospital, and fully informed consent was obtained from the patients.

## ADDITIONAL FILES

The following material is available online.

### Supplemental Material

**Fig. S1 (mSystems00339-25-s0001.tif).** Heat map of correlation between dominant bacteria and immune activation genes, immune suppressor genes, and chemokine receptors in NLNM and LNM groups.
**Fig. S2 (mSystems00339-25-s0002.tif).** Ranking the importance of gut microbiota associated with LNM to predict lymph node status in CRC patients.
**Legends (mSystems00339-25-s0003.docx).** Legends for supplemental figures and tables.
**Table S1 (mSystems00339-25-s0004.docx).** Results of LEfSe analysis.
**Table S2 (mSystems00339-25-s0005.docx).** KEGG pathways in the gut microbiota of CRC patients in LNM and NLNM.
**Table S3 (mSystems00339-25-s0006.pdf).** List of differential GO items and KEGG pathways of LNM and NLNM.
**Table S4 (mSystems00339-25-s0007.csv).** Association between LNM-associated GO and KEGG enrichment and the dominant gut microbiota in LNM and NLNM.

### Open Peer Review

**PEER REVIEW HISTORY (review-history.pdf).** An accounting of the reviewer comments and feedback.

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
