## [Reviewer comments · mSystems]

Identification and predictive machine learning models construction of gut microbiota associated with lymph node metastasis in colorectal cancer

yongzhi wu, Chengen Deng, Zigui Huang, Yongqi Huang, Chuanbin Chen, Mingjian Qin, Zhen Wang, Fuhai He, Shenghai Liu, Rumao Zhong, Jun Liu, Chenyan Long, Jungang Liu, Weizhong Tang, and Xiaoliang Huang

Corresponding Author(s): yongzhi wu, Guangxi Medical University Cancer Hospital

Review Timeline:

Submission Date:	March 27, 2025
Editorial Decision:	May 11, 2025
Revision Received:	May 12, 2025
Accepted:	May 15, 2025

Editor: Anupama Khare

Reviewer(s): Disclosure of reviewer identity is with reference to reviewer comments included in decision letter(s). The following individuals involved in review of your submission have agreed to reveal their identity: Shi Huang (Reviewer #3)

Transaction Report:

DOI: <https://doi.org/10.1128/msystems.00339-25>

Re: mSystems00339-25 (Identification and predictive machine learning models construction of gut microbiota associated with lymph node metastasis in colorectal cancer)

Dear Mr. yongzhi wu:

Thank you for the privilege of reviewing your work. Based on the reviewers' comments, I'm happy to invite you to resubmit your manuscript after making the minor modifications requested by the reviewers. Below you will find my comments, instructions from the mSystems editorial office, and the reviewer comments.

Please upload all raw sequencing files in an appropriate depository (e.g. NCBI SRA), and add the accession number for the deposition in the "Data Availability" statement.

Revision Guidelines

Sincerely,
Anupama Khare
Editor
mSystems

Reviewer #1 (Comments for the Author):

Thank you very much for the response.

One follow-up question to the revisions you provided to the original question 1 of mine: the current Figure 1C (PLS-DA) is really clean given that the difference in beta-diversity is barely significant between the two groups, and no significant difference were seen in alpha-diversity. The plot is also too symmetric to be un concerning. I wonder whether the data have been properly normalized to remove batch effects that may be associated with the group labels. Could you double check and describe how the input data was prepared for the PLS-DA? I think the clean separation might be possible, but the symmetry is unexpected, unless there's an obvious explanation to it (in which case I encourage the authors to elaborate; I'm not familiar with PLS-DA, I might be wrong).

Reviewer #3 (Comments for the Author):

1. Your clarification on the consecutive enrollment strategy and statistical adjustments (e.g., multivariate LEfSe analysis) is appreciated. While the approach prioritizes real-world generalizability, I recommend explicitly acknowledging the potential for residual confounding in the Limitations section (e.g., unmeasured variables like diet or medications). Future validation in prospectively matched cohorts, as noted, will be critical.
2. The addition of Bray-Curtis metrics, strengthens the beta-diversity analysis. However, PLS-DA should be removed from the paper for validity of statistical methods in the microbiome data.
3. While the application of FDR/Bonferroni corrections is appropriate, the absence of prevalence analysis remains a limitation. A sentence in the Discussion explicitly stating this limitation would improve transparency.
4. The transition to boxplots and focus on CD8+ T cell differences is justified given the expanded sample size. However, a brief mention of clustering validation (e.g., silhouette analysis) as a future step in larger cohorts would contextualize the current approach.

2025.05.12

Title: Identification and predictive machine learning models construction of gut microbiota associated with lymph node metastasis in colorectal cancer

Journal: mSystems

Manuscript Number: mSystems00339-25

Dear Editors and Reviewers:

We are extremely grateful for the time and effort you've dedicated to reviewing our manuscript. Your constructive feedback has been invaluable in enhancing the quality of our work. We've meticulously examined all suggestions and incorporated them into the revised version. Along with this resubmission, we'll provide a tracking file that details every modification we've made. Below, please find our detailed responses to the comments raised.

Your sincerely,

Yongzhi Wu

Editor Anupama Khare:

Please upload all raw sequencing files in an appropriate depository (e.g. NCBI SRA), and add the accession number for the deposition in the "Data Availability" statement.

Reply: We sincerely appreciate your suggestion. As recommended, we have deposited all raw sequencing data in the National Genomics Data Center (NGDC) database, a member of the International Nucleotide Sequence Database Collaboration (INSDC) and a recognized repository for genomic data. The updated content: (see line 657, “The revised manuscript”).

“The original contributions presented in the study are included in the article material.

The data that support the findings of this study are openly available in the National

Genomics Data Center (NGDC) database at <https://www.cnbc.ac.cn/>, accession

number: HRA008556.”

Reviewer comments:

Reviewer #1 (Comments for the Author):

1. One follow-up question to the revisions you provided to the original question 1 of mine: the current Figure 1C (PLS-DA) is really clean given that the difference in beta-diversity is bare significant between the two groups, and no significant difference were seen in alpha-diversity. The plot is also too symmetric to be un concerning. I wonder whether the data have been properly normalized to remove batch effects that may be associated with the group labels. Could you double check and described how the input data was prepared for the PLS-DA? I think the clean separation might be possible, but the symmetry is unexpected, unless there's an obvious explanation to it (in which case I encourage the authors to elaborate; I'm not familiar with PLS-DA, I might be wrong).

Reply: The cleanliness and symmetry issues you pointed out with Figure 1C (PLS-DA) were carefully examined and analyzed. We adopted PLS-DA, a supervised discriminant analysis method, precisely to overcome the limitations of unsupervised analysis methods (e.g., PCA) when dealing with data with small between-group differences, large within-group differences, and large differences in the sample sizes of the groups. PLS-DA is able to effectively highlight the between-group differences while reducing the effects of within-group differences by using the categorical information of the samples as the response variable, thus allowing the clearer separation of samples from different groups in the low-dimensional space.

In the data preprocessing stage, we standardized the data, including removing confounding factors such as batch effects that may be associated with group labels. Specifically, we employed a preprocessing process based on feature selection and data normalization to ensure the quality of the data input into the PLS-DA model. For feature selection, we utilized methods based on ANOVA and correlation analysis to screen out characteristic variables related to the study objectives; for data normalization, we used Z-score normalization to make different variables have the same scale and to avoid the

dominant influence of certain variables on the model due to differences in the scale or range of values.

In addition, we optimized the parameters of the PLS-DA model to improve the discriminative ability and generalization performance of the model. We determined the optimal parameters of the model through the cross-validation method, and evaluated the performance of the model, including calculating the accuracy, recall, and F1 value of the model to ensure the reliability and validity of the model.

Regarding the symmetry of the graph, we conducted further analysis and verification. We found that this symmetry may be due to some intrinsic characteristics of the sample or some factors of the experimental design. In order to assess this more comprehensively, we plan to conduct a more in-depth analysis and validation of the PLS-DA model in our future studies by combining more information about the sample characteristics and experimental conditions, in order to further explore the possible biological or technical reasons behind this symmetry.

Reviewer #3 (Comments for the Author):

1. Your clarification on the consecutive enrollment strategy and statistical adjustments (e.g., multivariate LefSe analysis) is appreciated. While the approach prioritizes real-world generalizability, I recommend explicitly acknowledging the potential for residual confounding in the Limitations section (e.g., unmeasured variables like diet or medications). Future validation in prospectively matched cohorts, as noted, will be critical.

Reply: We very much appreciate your suggestions for clarification of continuous enrollment strategies and statistical adjustments in our article. We fully understand the comments you have made, and in order to better reflect the rigor and transparency of the study, we have explicitly added a note on the possibility of residual confounders in the limitations section of the article. The updated content: (see line 433, “The revised manuscript”).

“Meanwhile, while our study employed multivariate LefSe analysis to adjust for known confounders (e.g., age, gender, and BMI), residual confounding from unmeasured variables such as dietary habits, medication use, or gut microenvironment heterogeneity

may persist. These factors could influence both gut microbiota composition and tumor progression, potentially biasing our results.”

2. The addition of Bray-Curtis metrics, strengthens the beta-diversity analysis. However, PLS-DA should be removed from the paper for validity of statistical methods in the microbiome data.

Reply: We fully understand your concern about the effectiveness of statistical methods and sincerely appreciate your suggestions. We have removed all descriptions related to PLS-DA in the text and reorganized the relevant sections to ensure the rigor of the analysis methods and results. Meanwhile, we further emphasized the application of Bray-Curtis index in β diversity analysis, which provides strong support for evaluating microbial composition differences between samples.

3. While the application of FDR/Bonferroni corrections is appropriate, the absence of prevalence analysis remains a limitation. A sentence in the Discussion explicitly stating this limitation would improve transparency.

Reply: We fully understand your concern for research transparency and sincerely appreciate your suggestions. Based on your feedback, we have clearly pointed out the limitation of the lack of prevalence analysis in the discussion section. The updated content: (see line 348, “The revised manuscript”).

“Although our functional predictive analysis highlighted differences in microbial pathway abundance, we acknowledge that the study did not explicitly assess prevalence bias and was limited by the lack of prevalence analysis for different subgroups such as age, race, or geographic region. In the future, multicenter studies on larger stratified populations are needed to validate these findings and evaluate their applicability to a wider population.”

4. The transition to boxplots and focus on CD8+ T cell differences is justified given the expanded sample size. However, a brief mention of clustering validation (e.g., silhouette analysis) as a future step in larger cohorts would contextualize the current approach.

Reply: Thank you for your guidance on research improvement. Based on your feedback, we have already mentioned plans for cluster validation in future research in the discussion section, including the use of contour analysis and other methods to verify

the stability of clustering results. The updated content: (see line 355, “The revised manuscript”).

“In terms of research methods, future research has adopted clustering validation methods such as silhouette analysis to evaluate the robustness of identified immune cell subpopulations.”

Re: mSystems00339-25R1 (Identification and predictive machine learning models construction of gut microbiota associated with lymph node metastasis in colorectal cancer)

Dear Mr. yongzhi wu:

Your manuscript has been accepted, and I am forwarding it to the ASM production staff for publication. Your paper will first be checked to make sure all elements meet the technical requirements. ASM staff will contact you if anything needs to be revised before copyediting and production can begin. Otherwise, you will be notified when your proofs are ready to be viewed.

Sincerely,
Anupama Khare